# A Novel Approach for the Production of Mildly Salted Duck Egg Using Ozonized Brine Salting

**DOI:** 10.3390/foods12112261

**Published:** 2023-06-03

**Authors:** Chantira Wongnen, Worawan Panpipat, Nisa Saelee, Saroat Rawdkuen, Lutz Grossmann, Manat Chaijan

**Affiliations:** 1Food Technology and Innovation Research Center of Excellence, School of Agricultural Technology and Food Industry, Walailak University, Nakhon Si Thammarat 80160, Thailand; chantira.wo@mail.wu.ac.th (C.W.); pworawan@wu.ac.th (W.P.); snisa@wu.ac.th (N.S.); 2Food Science and Technology Program, School of Agro-Industry, Mae Fah Luang University, Chiang Rai 57100, Thailand; saroat@mfu.ac.th; 3Department of Food Science, University of Massachusetts Amherst, 102 Holdsworth Way, Amherst, MA 01002, USA; lkgrossmann@umass.edu

**Keywords:** salted egg, ozone, albumen, protein oxidation, brining

## Abstract

Salted eggs are normally produced by treating fresh duck eggs with a high salt concentration in order to acquire distinctive features and excellent preservation capabilities as a result of a series of physicochemical changes. This method, however, induces a high salt content in the product. The goal of this research was to create a new way of producing mildly salted duck eggs using ozonized brine salting. The brine was made by dissolving NaCl (26% *w*/*v*) in water or ozonized water at a concentration of 50 ng ozone/mL (ozonized brine). Compared to brine, ozonized brine resulted in salted eggs with reduced ultimate salt levels in both albumen and yolk (*p* < 0.05). The Haugh unit of the salted eggs generated by ozonized brine was similar to that of the brine-made salted egg group (*p* > 0.05), but the salted egg produced by ozonized brine matured and solidified faster because the yolk index (0.62) was higher than that of the brine (0.55) (*p* < 0.05). The final pH of salted eggs generated with brine and ozonized brine was not different (*p* > 0.05). Regardless of the salting method, both salted eggs contained low TVB-N content (<10 mg/100 g). Ozonized brine increased the protein carbonyl content in salted albumen, which may be related to albumen protein aggregation and served as a salt diffusion barrier. However, after boiling the salted egg, the protein carbonyl level was comparable to that of fresh albumen. The TBARS levels of boiled salted albumen prepared with brine and ozonized brine were comparable (*p* > 0.05), and the value was extremely low (~0.1 mg MDA equivalent/kg). The TBARS value of the salted yolk prepared with brine was higher than that of the salted yolk prepared with ozonized brine (*p* < 0.05), and both salted yolks showed increased TBARS values after cooking (*p* < 0.05). The albumen and yolk components appeared to be altered similarly by both brine and ozonized brine, according to the FTIR spectra. Furthermore, the appearance and color of the yolk and albumen in salted eggs prepared with brine and ozonized brine were comparable. Boiled salted albumen produced with ozonized brine had a denser structure with fewer voids. This could be attributed to the final salted egg’s lower salt content and lower salt diffusion rate, which were likely caused by protein oxidation and, as a result, aggregation when ozonized brine was used.

## 1. Introduction

Eggs are an important component in the human diet and are acknowledged as palatable, functional, and rich in nutrients [1,2]. Eggs are a versatile food that contains essential amino acids (protein digestibility–corrected amino acid score (PDCAAS) of 1.00 for egg albumen protein), vitamins, and minerals (for example, vitamin A, folate, selenium, choline, phosphorus, and vitamin D, B12, riboflavin, biotin, and iodine) [1,2]. Eggs, on the other hand, are perishable, similar to raw meat, poultry, and fish, and are prone to contamination by certain microorganisms, e.g., *Salmonella* spp. [3]. Increased albumen pH, albumen thinning, and water evaporation through the shell are all well-known events that result in the deterioration of internal characteristics (albumen, yolk, weight, and pH) during storage [3]. Pore canals in eggshells allow the mass transfer, mostly carbon dioxide and moisture, resulting in changes in egg yolk and albumen, as well as weight loss [4] and the entry of contaminants, such as microorganisms and unpleasant odors under certain conditions [5]. To overcome these challenges, eggs can be immersed in salt solutions to preserve them, maintain their nutritional value, and extend their shelf-life [6].

Salted duck egg is a typical preserved egg product that is popular and considered a delicacy in Asia. The salted duck eggs are steamed or boiled and then peeled before eating. The two most frequent ways to make salted eggs are to immerse the eggs in saturated brine (brining method) or to cover the eggs with a salted earth paste (coating method) [7]. Sodium chloride (NaCl) inhibits the growth of microorganisms that cause deterioration, hence extending the shelf life [7]. The salted egg’s characteristics are mostly related to the egg white’s hydration and the yolk’s solidification after salting and cooking [8]. The brining or brine immersion technique is currently a typical processing method for the manufacture of salted eggs since it is a straightforward procedure with low manufacturing expenses and is utilized for mass production [9]. The process is typically regulated by adjusting the brine strength and incubation time [7]. Although the salting procedure provides the eggs with unique characteristics, it also introduces a significant quantity of salt into the eggs; for example, the salt concentration in egg whites can reach 7–10% after salting [10]. Reduced salt consumption and the development of salt alternatives have become major research subjects, and low-sodium foods are increasingly being advocated [10,11,12]. 

As they are a major egg product, there is a high interest in reducing the salt content of salted eggs. As a result, numerous techniques have been developed to manufacture reduced-sodium salted duck eggs, such as pulsed pressure, circulating water, and consecutive pickling with high and low concentrations of NaCl [13]. However, the flavor of these products may differ from the flavor of the eggs pickled via the original method [13]. As an alternative way of producing salted eggs, the use of ozone in brine, also referred to as “ozonized brine”, is a promising novel method. Because of its oxidizing potential, it has been hypothesized that ozone enhances the shelf life and promotes a delay in salt diffusion. 

Ozone (O_3_) is a gaseous substance that occurs naturally in low quantities (0.05 mg/L) in the stratosphere by the action of solar UV irradiation on oxygen. A trace quantity of ozone is also generated in the troposphere as a byproduct of photochemical reactions involving hydrocarbons, oxygen, and nitrogen emitted by automobiles, industries, forests, and volcanic activity [14,15]. Ozone technology is more efficient and cleaner than other traditional sanitizers, such as chlorine, and it is easier and less expensive to be produced industrially [14,15,16,17,18,19]. The US Food and Drug Administration has approved it as a potent oxidant and a safe disinfectant when used in direct contact with food [14,15]. Moreover, it is widely recognized as environmentally friendly [16]. Because of its antimicrobial efficacy, ozone has previously been used to inactivate microorganisms in drinking water [17]. Ozonated water is a powerful oxidant that reduces both spoilage and pathogenic bacteria [18]. Ozone damages bacteria by oxidizing cell walls and cytoplasmic membranes [19]. Furthermore, studies have shown that ozone treatment is effective in inactivating a wide range of bacteria, including *Staphylococcus aureus*, *Escherichia coli*, *Enterococcus faecalis*, and *Pseudomonas aeruginosa* [19]. Furthermore, ozone has been employed in the removal of off-flavors and as a bleaching agent in food processing, which helps to increase the whiteness of food products, such as surimi and other seafood [16].

In the context of egg quality and preservation, ozone has the potential to considerably improve the quality of egg products in terms of shelf life, sensory, and other aspects [16]. According to Yuceer et al. [20], ozone treatments result in improved egg quality for a longer period of time. Furthermore, it has been demonstrated that ozone–ultrasonic treatment helps to retain the internal quality of the egg and extend its shelf life [21]. Ozone concentrations of 2 and 4 ppm were found to be effective in preserving the interior quality and functional qualities of fresh eggs during storage. Ozone treatment has been utilized to help increase the shelf life of whole eggs as well as to improve the functionality of egg yolks, such as emulsifying activity [14,20,22]. The mild ozonation treatment (≤20 min) exposed more internal hydrophobic groups of egg yolk proteins and increased protein flexibility to improve the emulsifying capabilities of liquid egg yolk. The viscoelasticity of the egg yolk emulsions could be significantly improved by ozonation [22]. 

A fundamental scientific explanation of the mechanism of action on proteins during the ozone oxidation reaction has been established [14]. Several studies have demonstrated that the generated reactive oxygen species (ROS) after ozone treatment result in protein oxidation and accelerate the breakdown of other compounds [14]. By oxidizing lipids, large levels of ROS can further cause oxidative damage to proteins [23]. The chicken egg yolk gel was considerably enhanced, and emulsification performance was improved following ozonation [24]. At the same time, the changes in volatiles and fatty acids in egg yolk under different ozone treatment periods have been evaluated [14]. It was found that, after ozonation, the composition and concentration of volatile compounds, as well as the fatty acid content of the egg yolk, were considerably altered. The aldehyde concentration of the egg yolk was increased from 78% to 95% after 30 min of ozone treatment, but the relative level of dibutyl amine reduced from 1.5% to 0.0% [14]. However, ozone has not yet been used in the manufacture of salted eggs. Ozone can cause modifications in albumen proteins, which can then be cross-linked to produce structure-like aggregate layers to delay salt penetration and, therefore, obtain mildly salted eggs without microbial deterioration. Thus, the purpose of this study was to develop a new method for producing mildly salted duck eggs by utilizing ozonized brine salting. The egg quality indices, as well as the physicochemical and chemical features of salted eggs produced with ozonized brine, were compared to brine. 

## 2. Materials and Methods

### 2.1. Chemicals

All chemicals used in this work, such as silver nitrate, nitric acid, potassium ferrocyanide, trichloroacetic acid (TCA), potassium carbonate, thiobarbituric acid (TBA), and 2,4-dinitrophenylhydrazine (DNPH), were purchased from Sigma-Aldrich Co. (St. Louis, MO, USA).

### 2.2. Salt Treatment

Fresh duck eggs measuring 65 g to 75 g were purchased from a farm in Thasala, Nakhon Si Thammarat, Thailand, less than 2 days after laying. The eggs were washed with tap water, air-dried, and examined for cracks. The salted eggs were allocated into two groups, those treated with brine (26% *w*/*v* NaCl) and those treated with ozonized brine (26% *w*/*v* NaCl dissolved in ozonized water at a concentration of 50 ng ozone/mL). An ozone generator (Ozoner^®^-020, ProTechSci Co., Ltd., Chiang Mai, Thailand) with a maximum ozone output of 500 mg/h was used to produce the ozonized water. The ozone generator was only used to prepare ozonized water. The ozonized water was then used to dissolve the salt, and the ozonized brine was used to produce the salted eggs. The duck eggs were entirely submerged in the brine or ozonized brine at ambient temperature (27–29 °C) in sealed glass containers for up to 15 days at a ratio of eggs to brine of 1:3 (*w*/*v*). The salt content of manually separated albumen and yolk was measured every 5 days. On Day 15, salted eggs were recovered and analyzed in comparison to fresh eggs. To obtain cooked salted eggs, the salted eggs were boiled for 20 min and then cooled to room temperature with running tap water before being evaluated for quality characteristics.

### 2.3. Analyses

#### 2.3.1. Salt Content

Salt concentrations in albumen and yolk were measured using an AOAC method [25] on days 0, 5, 10, and 15 of salting. One gram of the sample was combined with 20 mL of 0.1 N AgNO_3_ and 10 mL of the concentrated HNO_3_. For 10 min, the mixture was slowly heated on a hot plate. Then, 5 mL of ferric alum indicator (FeNH_4_(SO_4_)_2_•12H_2_O) was added after cooling with running tap water. The mixture was titrated with standardized 0.1 N KSCN until the color became permanently light brown. The salt content was determined as follows:(1)$$Salt\;\left(\%\right)=\frac{5.8\times [\left(V1\times N1\right)-\left(V2\times N2\right)]}{W}$$
where 5.8 = conversion factor; V1 = volume of AgNO_3_ (mL); N1 = AgNO_3_ concentration (N); V2 = volume of KSCN (mL); N2 = KSCN concentration (N); and W = sample weight (g).

#### 2.3.2. Interior Quality

Haugh unit (HU) and yolk index, two interior quality indices of salted eggs (Day 15), were determined compared to fresh eggs [26]. The HU is an expression that relates egg weight and thick albumen height and is used to assess albumen quality. The higher the HU, the greater the albumen quality of the eggs [3]. The yolk index is a measure of the quality of the yolk [3].

The following formula was used to determine the HU:(2)$$\mathrm{HU}=100log\left(h-1.7G^{0.37}+7.6\right)$$
where G is the total egg’s mass in grams, and h is the thick albumen’s height in millimeters. The parameter (h) was calculated by averaging three measurements made with a digital caliper (CD-15CP, Mitutoya Ltd., London, UK) at various places of thick albumen spaced 10 mm apart from the yolk.

A digital caliper was also used to measure the yolk, and the yolk index was derived by dividing the yolk height by the yolk width. 

#### 2.3.3. The pH and Total Volatile Base Nitrogen (TVB-N)

The pH and TVB-N contents of albumen and yolk from raw fresh eggs and raw salted eggs produced with brine and ozonized brine (Day 15) were determined. The pH of 10 mL samples was measured using a pH meter (Cyberscan 500, Singapore). The TVB-N contents were measured via the method of Conway and Byrne [27]. The sample (2 g) was vigorously mixed with 8 mL of 4% TCA. The filtrate (1 mL) was placed in the outer ring after the mixture was filtered with Whatman No. 41 filter paper. The inner ring solution (1% boric acid with the Conway indicator) was pipetted into it. Then, K_2_CO_3_ (1 mL) was combined with the filtrate to start the reaction. The Conway unit was sealed and incubated for 60 min at 37 °C. After that, the inner ring solution was titrated with 0.02 N HCl until the green color changed to pink. The TVB-N content is presented in mg/100 g.

#### 2.3.4. Protein and Lipid Oxidation

Protein carbonyl content was utilized to detect protein oxidation, and thiobarbituric acid reactive substances (TBARS) were used to determine lipid oxidation in raw and boiled salted duck eggs compared to raw and boiled fresh eggs [28].

For protein carbonyl, 2 mL of 10 mM DNPH in 2 M HCl was reacted with 0.5 mL of 4 mg/mL protein solution prepared from albumen and yolk for 1 h at room temperature (27–29 °C). After incubation, 2 mL of 20% (*w*/*v*) TCA was added to the precipitate protein. The resulting pellet was washed repeatedly with 4 mL of ethanol:ethylacetate (1:1, *v*/*v*), blow-dried, and placed in 1.5 mL of 0.6 M guanidine hydrochloride in 20 mM potassium phosphate (pH 2.3) to separate unreacted DNPH. The protein absorption was measured at 370 nm, and the protein carbonyl content was calculated using a molar absorptivity of 22,400 M^−1^cm^−1^.

For TBARS, the sample (0.5 g) was homogenized in 2.5 mL of a TBARS solution (0.375% TBA, 15% TCA, and 0.25 N HCl). The homogenate was heated in a boiling water bath (95 °C) for 10 min to create a pink color, cooled with running tap water, then centrifuged at 3600× *g* at 25 °C for 20 min. Following that, the absorbance of the supernatant was measured at 532 nm. A standard curve was created using 1,1,3,3-tetramethoxypropane at values ranging from 0 to 10 ppm. TBARS was reported as mg malondialdehyde (MDA) equivalent/kg sample.

#### 2.3.5. Fourier Transform Infrared (FTIR) Spectra

The FTIR spectra of freeze-dried samples were collected using a Bruker INVENIO-S FTIR spectrometer (Bruker Co., Ettlingen, Germany) fitted with an attenuated total reflection (ATR) diamond crystal cell. A 32-scan with a resolution of 4 cm^−1^ was used to calculate the IR absorption in the 400–4000 cm^−1^ range. OPSU 8.5 software (Copyright^©^ Bruker Optik GmbH 2020, Ettlingen, Germany) was used to analyze the data [29].

#### 2.3.6. Color

The lightness (*L**), redness/greenness (*a**), and yellowness/blueness (*b**) of boiled salted eggs and boiled fresh eggs were measured with a colorimeter in triplicate using a calibrated Hunterlab ColorFlex^®^EZ instrument (10° standard observers, illuminant D65, Hunter Assoc. Laboratory, Reston, VA, USA). The following equations were used to compute the albumen whiteness value and the yolk redness index [30]:(3)$$Whiteness=100-\sqrt{{[(100-L^{*}}^{2})+{a^{*}}^{2}+{b^{*}}^{2}]}$$
(4)$$Redness\;index=\frac{a^{*}}{b^{*}}$$

#### 2.3.7. Microstructure

The microstructures of boiled salted eggs were examined in accordance with the technique provided by Somjid et al. [31]. After being sliced into 2–3 mm cubes, the samples were fixed for 2 h at room temperature, with 2.5% (*v*/*v*) glutaraldehyde in 0.2 M phosphate buffer (pH 7.2), and then washed twice with 0.1 M phosphate buffer. The samples were first fixed with 1% osmium tetroxide, then further washed in 0.1 M phosphate buffer and deionized water, respectively. The samples were sequentially dehydrated in ethanol at concentrations of 50, 70, 80, 90, and 100% (*v*/*v*), followed by a critical point of drying with CO_2_ as the transition fluid. The gold was sputter-coated after being applied to dried samples placed on a bronze stub. The samples were scanned using an SEM (GeminiSEM; Carl Zeiss Microscopy, Berlin, Germany) with 2 kV of acceleration and 1000–10,000 times magnification.

### 2.4. Statistical Analysis

All data were analyzed through a one-way ANOVA using SPSS 16.0 for Windows (SPSS Inc., Chicago, IL, USA). A comparison of means was carried out using Duncan’s multiple-range test. All analyses were performed in triplicate.

## 3. Results and Discussion

### 3.1. Changes in Salt Content

Initially, the salt concentration during salting in albumen and yolk made with brine and ozonized brine was monitored in order to comprehend the impact of ozonation on salt diffusivity. Figure 1 shows the changes in salt concentration (as NaCl) of duck egg albumen and yolk during salting using brine and ozonized brine. Salt concentrations of both albumen and yolk increased after salting in both brine solutions. Following salting with both types of brine solutions, albumen had a higher salt concentration than yolk (*p* < 0.05). In comparison to brine, ozonized brine produced salted eggs with lower ultimate salt levels in both albumen and yolk (*p* < 0.05). Salted albumen had a final salt concentration of 5.45% in brine compared to 4.18% in ozonized brine. For the salted yolk, the final salt contents of the brine and ozonized brine were 1.84% and 1.43%, respectively. 

This finding could be attributed to the fact that the ozonized brine may lead to protein modifications that slow down the diffusion by forming a denser network similar to how aggregated proteins hindered salt diffusion into the interior of the egg. This effect was more evident in the albumen part. In contrast, the salt content in the yolk initially increased faster when the brine was ozone-treated compared to the untreated brine solution. This initial high uptake of salt then leveled off and remained almost constant after 5 days until the end of the process. The decreased salt level in the yolk of the ozonized brine treatment on Day 15 was likely caused by the lower rate of salt diffusivity, which could have been influenced by the structural changes in the yolk and the vitelline membrane that eventually prevented the salt penetration, as indicated above. 

In general, our results show a similar trend compared to what was found by other authors [32,33]. Kaewmanee et al. [32] found that as salting time increased, the moisture content of both albumen and yolk reduced steadily, with simultaneous increases in salt and ash content. After 14 days of salting using the coating method, the salted duck egg’s albumen and yolk had salt contents of roughly 7% and 1%, respectively [32]. Overall, these results show that mildly salted duck eggs can be produced using ozonized brine.

### 3.2. Interior Quality Indices

Salting and ozone treatment can have a multitude of effects on the quality parameters of both the egg albumen and egg yolk. One critical parameter is the HU, which was measured in several experiments. According to Yuceer and Caner [26], greater HU values indicate higher protein functionality. HU over 72 is considered an AA grade. HU between 71 and 60 is regarded as an A grade. B grade is assigned to HU between 59 and 31, and C grade is given to HU below 30 [26].

The HU in the salted egg with brine (62.47) and ozonized brine (63.58) was lower than in the fresh egg (92.71; *p* < 0.05) after 15 days of salting (Figure 2a). This effect was due to the fact that egg white liquefies after salting [7]. However, the ozonized brine-produced salted eggs had a similar HU to the brine-produced salted egg group (*p* > 0.05).

The ratio of yolk height to yolk diameter, known as the yolk index of salted eggs, provides information about the degree of hardening of the salted egg yolk. The yolk index of a fresh, high-quality egg is about 0.45 [26]. According to Yuan et al. [34], the salted egg is ready to be harvested when the yolk index is close to 1.0. In this experiment, the ozonized brine-salted egg had a slightly higher yolk index (0.62) than the brine-salted egg (0.55) and fresh egg (0.47), respectively (*p* < 0.05; Figure 2b). As a result, ozonized brine-salted eggs had greater yolk index quality than brine-salted eggs, and they might be ready to be harvested sooner. In general, the egg yolk gradually hardens and solidifies while being salted. It is well known that adding salt causes the desired characteristics of salted egg yolks, including changing the color intensity, inducing a sandy–oily texture, and creating a pleasant aroma [13]. In conclusion, these measurements show that, in comparison to brine, ozonized brine causes the yolk to solidify to a greater extent.

### 3.3. The pH and TVB-N

Figure 3a illustrates the pH values of fresh duck egg albumen and yolk as well as salted egg albumen and yolk made with brine and ozonized brine. Fresh albumen had a pH of 8.7, but following salting with both brines, the pH was lowered to about 7.7. After salting with both brines, the pH of the yolk was slightly raised from 5.6 to 6.0. Salted eggs produced with brine and ozonized brine did not differ in their final pH for the albumen or yolk (*p* > 0.05). According to Yuceer and Caner [26], the balance between dissolved carbon dioxide, bicarbonate ions, carbonate ions, and proteins determines the pH of eggs. The breakdown of carbonic acid in the yolk may release carbon dioxide, changing the bicarbonate buffer system and raising the pH of the yolk [35]. Additionally, the carbonic acid may pass from the yolk to the albumen and may become trapped in the protein cross-links that make up the albumen structure. Some coating substances, including chitosan and mineral oil, have been shown to be efficient barriers against the leakage of carbon dioxide and other gases through the eggshell, assisting in maintaining a low pH in the albumen [26,35,36].

Fresh albumen and yolk showed a value for the TVB-N of 2 mg/100 g. While the TVB-N of albumen remained unchanged after brine salting, the TVB-N of yolk increased two-fold (Figure 3b). The albumen and yolk TVB-N levels were elevated by almost three times for salted eggs made with ozonized brine. Proteins may be chemically altered by ozonized brine, and TVB-N compounds may be formed as a co-product. TVB-N is an indicator that is regularly used to assess the freshness of fish and fish products but also has recently been employed for other protein-rich foods, such as meat and meat products. This is due to its favorable correlation with the growth of microorganisms and the actions of proteolytic enzymes, both of which are significant mechanisms for reducing the shelf-life of foods [37]. According to Bekhit et al. [38], these biochemical and microbiological activities specifically contribute to the production of ammonia, biogenic amines, as well as other products of amino acid deamination and decarboxylation that are collectively referred to as “TVB-N”. The National Food Safety Standard of China [39] defines the limit of freshness for beef or cattle meat products as TVB-N < 15 mg/100 g. There is no TVB-N value specified for salted eggs. A solid generalization is that products with low TVB-N values are likely to be fresh, whereas those with high TVB-N values are likely to be spoiled. From the results, all salted eggs, regardless of the method of salting, contained less TVB-N (<10 mg/100 g) than the proposed acceptable level for fresh food [38,39]. Therefore, ozonized brine can be used to produce the salted egg with good quality without any detrimental effect on flavor reversion regarding the formation of TVB-N.

### 3.4. Protein and Lipid Oxidations

Ozone’s oxidative characteristics can induce egg protein and lipid oxidation. Figure 4a,b, respectively, shows the protein carbonyl contents of the albumen and yolk of fresh and salted duck eggs. Regardless of salting, the protein carbonyl content in the raw egg was higher than in the boiled egg for the albumen (Figure 4a). This was surprising because it was expected that the protein oxidation endproducts would be low in fresh eggs and increase over time. This might be due to the development of adducts as well as certain undetected volatile carbonyl compounds that resulted from heating. It is interesting to note that ozonized brine caused the salted albumen to have greater protein carbonyl levels, which may be related to the aggregation of albumen proteins that served as a barrier for salt diffusion. In comparison to using typical brine, the extent of salt penetration was, therefore, lowered. However, once the salted egg was boiled, the protein carbonyl content decreased significantly and was similar to fresh raw eggs. 

In the case of the yolk (Figure 4b), fresh yolk had a lower protein carbonyl level than salted eggs made with both brine and ozonized brine (*p* < 0.05). The boiled salted egg yolk made with ozonized brine had the highest protein carbonyl concentration. In contrast to albumen, the salted yolk made with ozonized brine had a higher post-boil protein carbonyl concentration. This was most likely caused by the more reactive carbonyl groups in the yolk than in the albumen, and these groups may undergo more extensive modifications during ozonized brine salting. It is well known that highly reactive, protein-bound, lipid oxidation-derived aldehydes, such as malondialdehyde, spontaneously attack non-lipid substrates (mostly proteins) during lipid oxidation [40]. As yolk contains a higher lipid content, it can be assumed that lipid oxidation is increased in this fraction, which, in turn, can foster protein oxidation. Moreover, reactive amino acids, including cysteine, tryptophan, methionine, tyrosine, and histidine in lysozyme, have been shown to be susceptible to oxidative modification by ozone [41].

To examine the extent of lipid oxidation of salted eggs, the albumen and yolk TBARS levels of raw and boiled fresh eggs (Day 0) were compared to raw and boiled salted eggs produced by brine and ozonized brine (Day 15). In general, lipids can be classified as not oxidized (TBARS value < 1.5 mg MDA/kg), moderately oxidized (1.6 < TBARS value < 3.6), or oxidized (TBARS value > 3.7), according to Larouche et al. [42]. In comparison to the yolk (Figure 5b), albumen (Figure 5a) contained lower amounts of TBARS. Fresh albumen had a lower TBARS value than raw salted albumen, regardless of boiling or salting (*p* < 0.05). Additionally, upon cooking in all treatments, the TBARS values increased (*p* < 0.05). The TBARS values of boiled salted albumen made with brine and ozonized brine were similar (*p* > 0.05), but the overall value was very low (~0.1 mg MDA equivalent/kg) and well below the undesirable level (>3.7 mg MDA equivalent/kg) [42].

The fresh egg was found to have the lowest TBARS levels for the yolk (~0.2 mg MDA equivalent/kg) in both raw and boiled forms (*p* < 0.05). The salted yolk prepared with brine had a greater TBARS value than that made with ozonized brine (*p* < 0.05), and both salted yolks had a rise in TBARS after cooking (*p* < 0.05). The increase in TBARS following cooking was typically caused by the oxidation of lipids caused by heating. Nevertheless, in all treatments, TBARS values were still within the acceptable range. It has been proposed that the ozonation of proteins and unsaturated lipids occurs concurrently and competitively, but the protein was found to be more vulnerable to oxidation than lipid in a study using glycophorin in solution and in lipid vesicles exposed to ozone [43]. It was claimed that ozonized brine appeared to induce protein oxidation to a desired degree. This was a benefit of using ozonized brine to make mildly salted eggs because the modified proteins (such as cross-linked proteins and aggregates) may act as a barrier to slow the diffusion of salt. 

### 3.5. FTIR Spectra

It has been demonstrated that FTIR is well adapted for monitoring changes to egg protein secondary structure brought on by raising temperature, freeze- and spray-drying, or varying ionic strength [44]. For this reason, the aim of the following measurements was to elucidate chemical changes in the protein and the lipid fraction using FTIR. 

In the case of albumen (Figure 6a), the spectra of all samples were practically identical, indicating that the same protein functional groups were discovered. Amide I, amide II, amide III, amide A, and amide B were the main bands in the FTIR spectra. The most intriguing band for the investigation of the secondary structure of proteins is the amide I band (1700–1600 cm^−1^) brought on by the stretching vibration of C=O [44]. According to Chaijan and Panpipat [29], amides I, II, and III have wavenumbers of 1646 cm^−1^, 1554 cm^−1^, and 1240 cm^−1^, respectively, for the vibration of C-O stretching, N-H group bending, and C-N group stretching. Doyle et al. [45] reported the N-H stretching vibration of amide A at 3416 cm^−1^, and Nagarajan et al. [46] measured the CH stretching vibration of amide B at 2958 cm^−1^. The FTIR spectra matched those that Guo et al. [47] found in the albumen of hen eggs. Additionally, it matched the duck albumen FTIR spectra reported by Ganezan et al. [48] that were located in the amide band region, specifically amide I (1600–1700 cm^−1^), amide II (1500–1600 cm^−1^), amide III (1200–1300 cm^−1^), amide A (3293–3306 cm^−1^), and amide B (2920–2922 cm^−1^). It seems that salting with either brine or ozonized brine did not significantly chemically modify albumen proteins.

In the case of yolk (Figure 6b), peaks in the same region at 2000–500 cm^−1^, as reported in albumen, were present with some exceptions. For example, the peak at 1744 cm^−1^ was noticeably visible. The ester linkages between glycerol and fatty acids are what is generally believed to cause this strong carbonyl peak to appear at 1744 cm^−1^ [49]. The peak around 1751 cm^−1^ of –C=O stretching vibration was reported to be the characteristic absorption peak of egg yolk lipids [44]. Additionally, the sharp peaks representing fatty acids were located between 2900 and 2800 cm^−1^. Due to the abundance of CH_2_ and CH_3_ groups in the fatty acids, the right part of the spectrum in the yolk is dominated by a series of aliphatic vibrations centered between 3500 and 3000 cm^−1^. Since yolk is a source of lipids, stretch vibration of CH bonds results in strong absorption at 2926 and 2855 cm^−1^. The OH stretching zone, or the absorbance at 3800–3100 cm^−1^ in the FTIR spectra, was described by Van de Voort et al. [50]. Due to their characteristic -OO-H stretching vibrations, hydroperoxide molecules exhibit characteristic absorption bands between 3600 and 3400 cm^−1^. Regardless of the type of brine, salted yolks were shown to have a higher absorbance band in this area than fresh yolks, indicating the development of hydroperoxide. Consequently, according to FTIR measurements, the yolk component appeared to be affected similarly by both brine and ozonized brine.

### 3.6. Appearance and Color

It is already known that salting eggs causes a considerable change in the appearance of raw eggs. As customers are used to the specific color of salted eggs, the aim of these analyses was to reveal whether ozonized brine resulted in a similar appearance. The salted yolks obtained by pickling in both brine and ozonized brine had a more spherical form and a more vibrant orange hue than the fresh yolk (Figure 7). Compared to the fresh egg, the salted (brine and ozonized brine) albumens were more turbid and had a lower viscosity. The reduced viscosity of salted albumen may result from less protein repulsion caused by higher ionic strength with increasing salt addition. The salting effect was responsible for both the reversal of color and the change in texture. Similarly, the appearance and color of the yolk and albumen were the same in salted eggs prepared with brine and ozonized brine, indicating the potential use of ozonized brine for producing mildly salted eggs without any effects on the color and appearance of the products. 

*L***a***b** measurements were carried out to further elucidate the color differences. In Figure 8, the whiteness of the albumen and the redness index of the yolk were measured as the color criteria. In terms of the whiteness of the albumen (Figure 8a), boiled fresh albumen was whiter than boiled salted albumen (*p* < 0.05), while the whiteness of the boiled salted albumens made with brine and ozonized brine was similar (*p* > 0.05). So, the ozonized brine had no adverse effects on the whiteness of the boiled salted albumen. The probable Maillard reaction during salting and cooking was likely the cause of a decrease in the whiteness of salted eggs compared to fresh eggs [8,44]. 

Regardless of the brine solution, the boiled salted egg had a greater redness index than the boiled fresh egg (*p* < 0.05) for the yolk redness index (Figure 8b). The loss of water in the yolk through osmosis during salting could be the principal reason why the pigment concentration was higher and could result in a higher redness in both salted yolks. This has previously been proposed as the mechanism that was responsible for the color change in salted duck egg yolk [8,44]. Overall, the redness index of yolk was similar in brine and ozonized brine. So, making mildly salted eggs with ozonized brine is possible without compromising the end product’s color characteristics.

### 3.7. Microstructure

The microstructures of boiled salted albumen and yolk prepared using brine and ozonized brine are shown in Figure 9. Generally, the viscosity, surface hydrophobicity, secondary structure, and microstructure of the egg albumens are significantly influenced by the addition of salt and, consequently, the ionic strength. Changes in these characteristics either directly or indirectly influence the hydration of egg proteins, whereas heat treatment causes protein aggregation and changes in secondary structure [51]. 

For egg yolks, gel formation is facilitated by the pickling process, which causes the moisture content of the egg yolk to decrease while increasing its viscosity and hardness [51]. This process also encourages the formation of disulfide bonds and boosts intermolecular or intramolecular connections [51]. The main variables influencing the development of the yolk gel are salt content, diffusion rate, and oil exudation [51]. According to Xu et al. [10], the increased chemical bonding in the yolk caused by the action of high salt causes the yolk to transform from spherical to irregular polyhedrons and become more densely organized, and the yolk oil is released in the spaces between the yolk spheres.

As indicated, boiled salted albumen made with ozonized brine had a denser, more compact structure with fewer pores. Ozone is a powerful oxidizing agent that may strengthen disulfide bonds. It has been suggested that ozone increases the density of the albumen gel network structure [52]. It has been proposed that ozone’s oxidative characteristics can cause the protein’s structure to change and the amino acid residues to oxidize. When the sulfhydryl groups are exposed, it can lead to modifications that are advantageous for gel formation. Additionally, during gel formation, carbonyl groups may establish covalent bonds with amino groups, improving protein interaction and, as a result, the performance of the gel [53]. This may be related to the difference in the final salt content, which was lower in the eggs incubated in ozone-treated brine (Figure 1). Here, the salt diffusion rate was lower, which was likely caused by protein oxidation and, consequently, aggregation when ozonized brine was utilized. When examining the yolk granules, it was evident that those made with ozonized brine had smoother surfaces than those made with brine. The number of holes and porousness in the salted yolk granules made with ozonized brine was also lower than in those made with brine. This fact demonstrated that albumen and yolk components tended to aggregate during ozonized brine salting, which was connected to a reduction in salt diffusivity as a result of the barrier effect of aggregated networks.

## 4. Conclusions

The traditional process of producing salted eggs involves treating fresh duck eggs with high salt concentration, which results in specific characteristics and preservation capabilities but also a high salt content that can result in high salt consumption. The goal of this study was to create a new method of producing salted duck eggs with a lower final salt content utilizing ozonized brine salting. When compared to brine, salted eggs prepared with ozonized brine showed lower salt levels in both albumen and yolk. Both groups’ HU were similar; however, the salted eggs generated by ozonized brine matured and solidified faster. It appeared that ozonized brine fostered protein oxidation but reduced lipid oxidation compared to regular brine treatment. Both groups had the equivalent yolk and albumen appearance, color, and structure. According to our results, ozonized brine salting can produce duck eggs with a lower salt content while retaining typical salting egg quality parameters. The postulated mechanism derived from this result was that the ozonized brine might cause protein modification in the way that aggregated proteins impeded salt migration into the egg’s interior. However, sensory evaluation, which is an important quality feature, can be undertaken further in the future to ensure the effectiveness of this approach.

## Figures and Tables

**Figure 1 foods-12-02261-f001:**
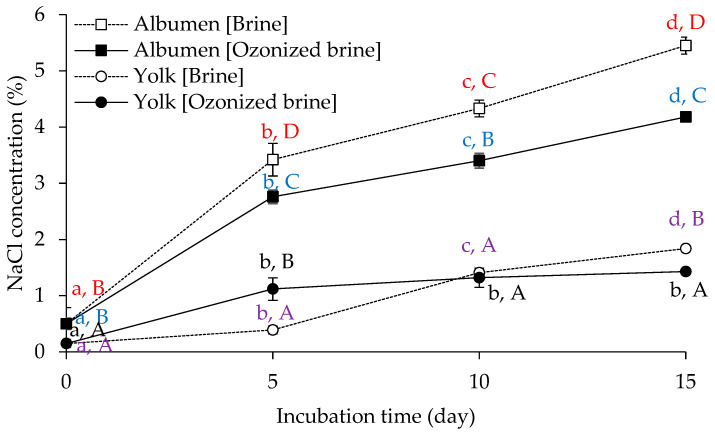
Changes in NaCl concentration of albumen and yolk of duck egg during salting using brine and ozonized brine. Bars represent standard deviation from triplicate determinations. Different lowercase letters on bars within the same sample indicate significant differences (*p* < 0.05). Different uppercase letters on bars within the same incubation time indicate significant differences (*p* < 0.05).

**Figure 2 foods-12-02261-f002:**
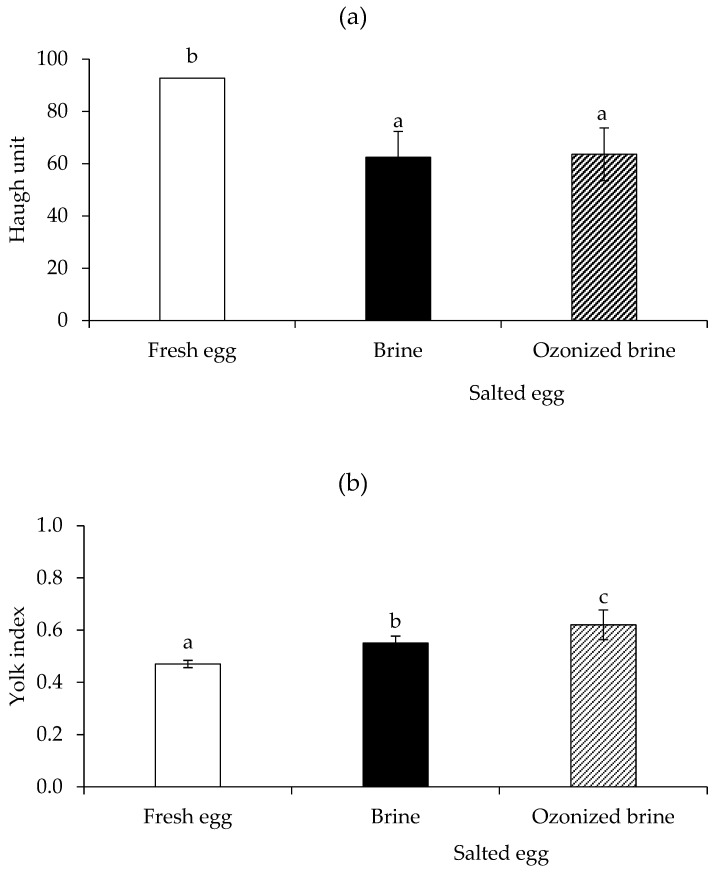
Haugh unit (**a**) and yolk index (**b**) of fresh eggs (Day 0) in comparison with salted eggs produced by brine and ozonized brine (Day 15). Bars represent standard deviation from triplicate determinations. Different letters on the bars indicate significant differences (*p* < 0.05).

**Figure 3 foods-12-02261-f003:**
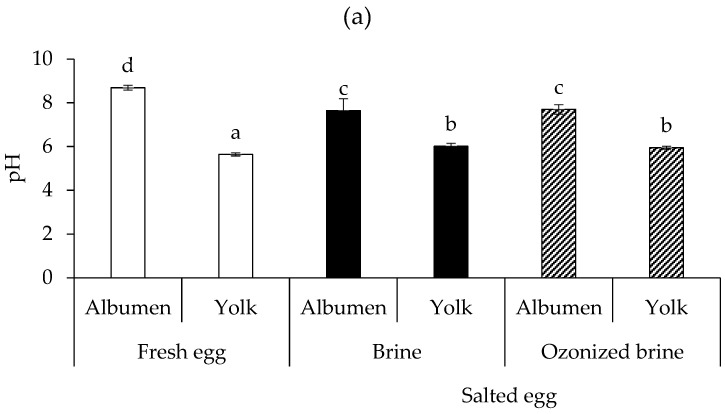
pH (**a**) and TVB-N contents (**b**) of albumen and yolk from raw fresh eggs (Day 0) and raw salted eggs produced by brine and ozonized brine (Day 15). Bars represent standard deviation from triplicate determinations. Different letters on the bars indicate significant differences (*p* < 0.05).

**Figure 4 foods-12-02261-f004:**
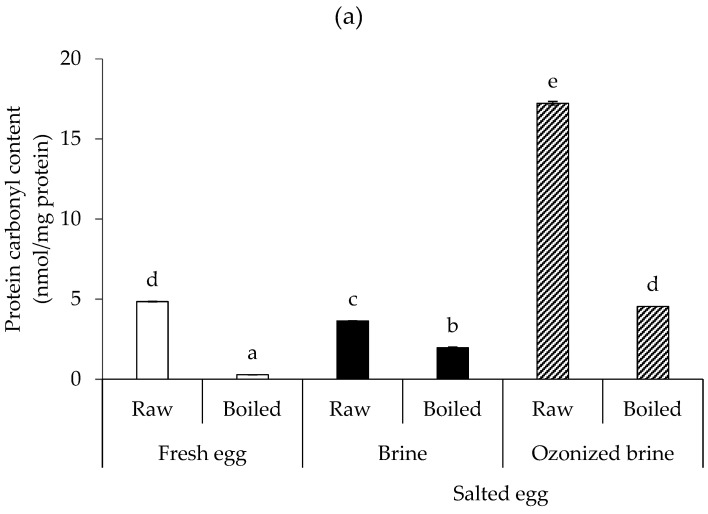
Protein carbonyl contents of albumen (**a**) and yolk (**b**) of raw and boiled fresh eggs (Day 0) in comparison with raw and boiled salted eggs produced by brine and ozonized brine (Day 15). Bars represent standard deviation from triplicate determinations. Different letters on the bars indicate significant differences (*p* < 0.05).

**Figure 5 foods-12-02261-f005:**
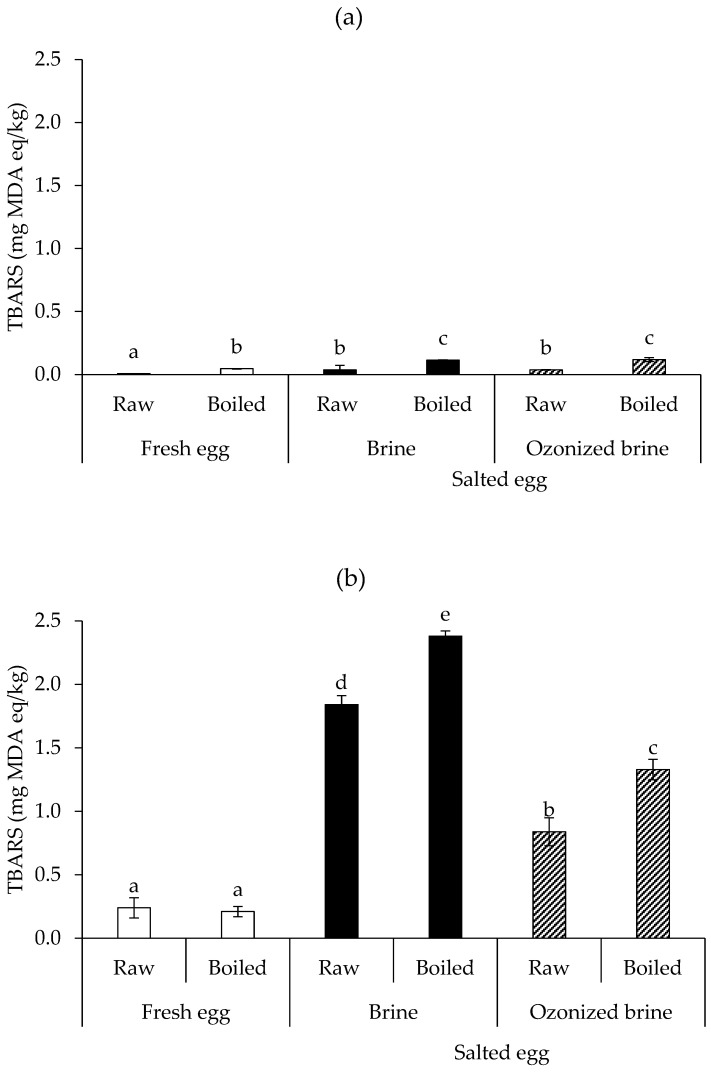
TBARS contents of albumen (**a**) and yolk (**b**) of raw and boiled fresh eggs (Day 0) in comparison with raw and boiled salted eggs produced by brine and ozonized brine (Day 15). Bars represent standard deviation from triplicate determinations. Different letters on the bars indicate significant differences (*p* < 0.05).

**Figure 6 foods-12-02261-f006:**
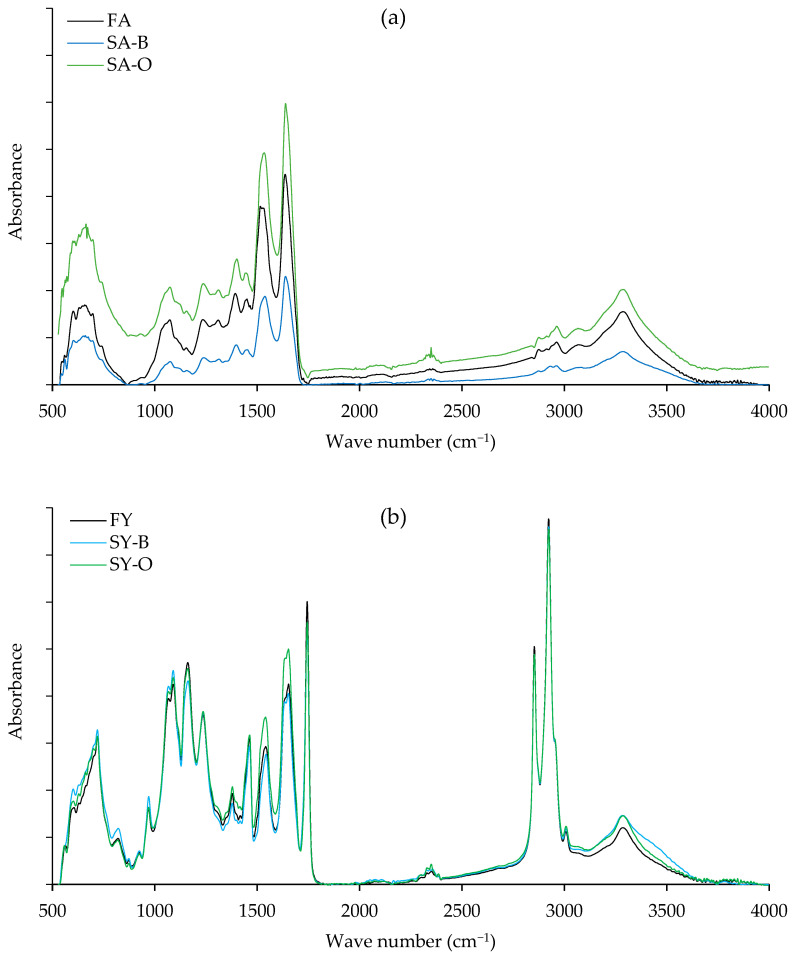
FTIR-spectra of albumen (**a**) and yolk (**b**) of boiled fresh eggs (Day 0) in comparison with boiled salted eggs produced by brine and ozonized brine. FA = fresh albumen; SA-B = salted albumen prepared using brine; SA-O = salted albumen prepared using ozonized brine; FY = fresh yolk; SY-B = salted yolk prepared using brine, and SY-O = salted yolk prepared using ozonized brine.

**Figure 7 foods-12-02261-f007:**
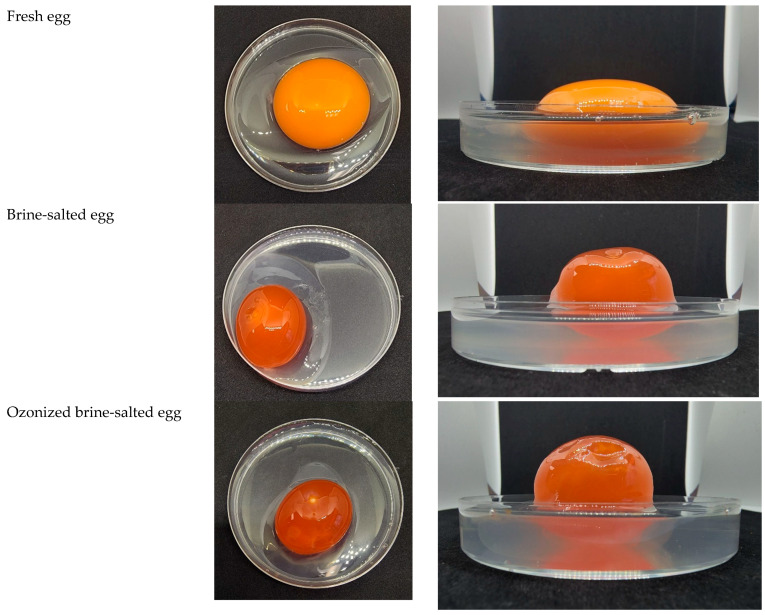
Appearance of fresh eggs and salted eggs (Day 15) made with brine and ozonized brine.

**Figure 8 foods-12-02261-f008:**
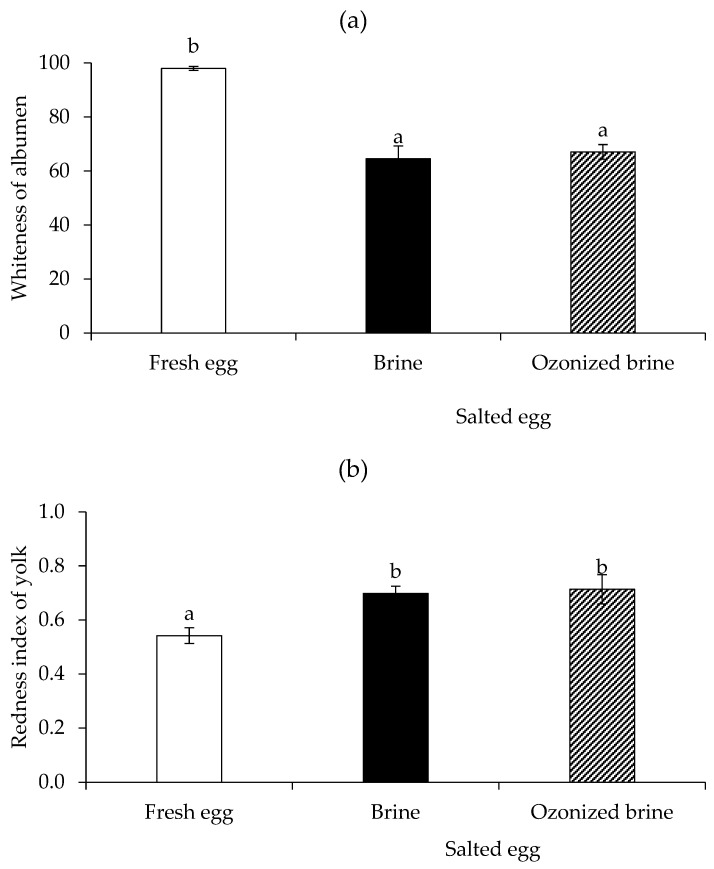
Whiteness of albumen (**a**) and redness index of yolk (**b**) of boiled fresh eggs (Day 0) in comparison with boiled salted eggs produced by brine and ozonized brine (Day 15). Bars represent standard deviation from triplicate determinations. Different letters on the bars indicate significant differences (*p* < 0.05).

**Figure 9 foods-12-02261-f009:**
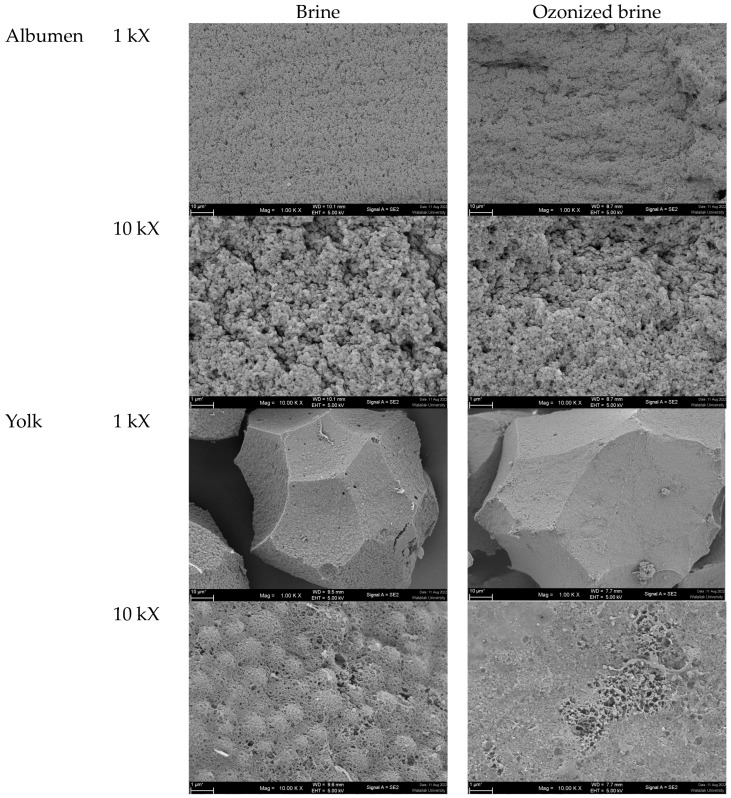
Microstructure boiled salted albumen and yolk prepared with brine and ozonized brine with 2 kV of acceleration and 1000–10,000 times magnification.

## Data Availability

The data used to support the findings of this study can be made available by the corresponding author upon request.

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
