# Peer review of "A Novel Approach for the Production of Mildly Salted Duck Egg Using Ozonized Brine Salting"

_foods, 2023, doi:10.3390/foods12112261_

Round 1

Reviewer 1 Report

The manuscript focuses on using ozonised brine salting method to produce salted eggs with lower salt content. The idea of using ozonised brine is interesting. The manuscript is well-written, organised, and easy to follow.

Some general comments on the study:

1.     Line 16: It is weird to talk about health concerns when linked to consuming salted eggs. Although the produced salted eggs contain less salt (Fig. 1), there is no clear indication that consuming salted eggs produced using the proposed method may address health concerns. Since this manuscript does not cover this area, I suggest authors remove statements related to health concerns (e.g. Line 15-16, 67-71). Authors may focus on the other advantage(s) of using ozonised brine salting method, but not related to health.

2.     Why do authors only analyse TBARS? How about the peroxide value?

3.     Sensory should be conducted to evaluate the consumers’ acceptance of the salted eggs produced using ozonised brine salting method. Sensory evaluation is one of the important quality aspects that is missing in this study.

4.     Include the limitation(s) of the study at the end of your conclusion section.

Some specific comments on the manuscript:

1.     Line 72: How do authors confirm that salted egg is a major egg product? Any statistical data to support this claim?

2.     Line 141-143: The preparation for salted eggs using ozonised brine salting is unclear. Do you mean that the ozone generator is only used to prepare ozonised water? The ozonised water is then used to dissolve the salt, and the ozonised brine is used to prepare the salted eggs. The ozone generator is not used throughout the salting duration (15 days). Please ensure the preparation method is explained in detail so that readers can reproduce the method.

3.     Line 154: AgNO3

4.     Eq. 1: Please check your equation. Also, indicate the meaning of “5.8” in the equation.

5.     Fig. 1: Please provide statistical analyses on the data points. Are they significantly different?

6.     Line 413: Please check the superscript.

7.     Fig. 7: Authors should also show the salted egg yolk’s cross-section or interior appearance. Solidification starts from the outer layer. Hence, showing the cross-section of the salted egg yolk is an important criterion to determine the salted egg’s readiness. 

8.     Line 533: Without sensory evaluation, the acceptance is merely related to visual appearance. Please indicate this in the manuscript.

I am satisfied with the quality.

Author Response

Reviewer#1

The manuscript focuses on using ozonised brine salting method to produce salted eggs with lower salt content. The idea of using ozonised brine is interesting. The manuscript is well-written, organised, and easy to follow.

Some general comments on the study:

  1. Line 16: It is weird to talk about health concerns when linked to consuming salted eggs. Although the produced salted eggs contain less salt (Fig. 1), there is no clear indication that consuming salted eggs produced using the proposed method may address health concerns. Since this manuscript does not cover this area, I suggest authors remove statements related to health concerns (e.g. Line 15-16, 67-71). Authors may focus on the other advantage(s) of using ozonised brine salting method, but not related to health.

Ans: Thank you very much. The statements related to health concerns in Line 15-16 and Line 67-71 were removed.

  1. Why do authors only analyse TBARS? How about the peroxide value?

Ans: In general, PV and TBARS are lipid oxidation indices that can be utilized to monitor the progression of lipid oxidation throughout food preparation and storage. PV is a measure of primary lipid oxidation product, which decomposes easily to create secondary lipid oxidation product. Only TBARS was used in this investigation since it is a biomarker of the accumulation of secondary lipid oxidation products such as aldehyde, which is linked to the rancid off-odor of salted eggs. Furthermore, the permissible level of TBARS in food products has been thoroughly proposed and can be applied to a wide range of food products, including salted egg.

  1. Sensory should be conducted to evaluate the consumers’ acceptance of the salted eggs produced using ozonised brine salting method. Sensory evaluation is one of the important quality aspects that is missing in this study.

Ans: Thank you very much for your helpful recommendations. For our next study, we will conduct a sensory test. It is undeniably true that sensory is a significant quality factor in judging product quality. However, the focus of this research is on understanding the mechanisms underlying the physicochemical and microstructural features. We attempted to monitor chemical indices such as TVB-N and TBARS in order to correlate with off-odor/off-flavor components that can be linked to sensory acceptability. We implied that if those chemical indices were within the permissible range, sensory acceptance should be satisfactory. To avoid dramatization, a mention concerning the sensory characteristic was removed from the text.

  1. Include the limitation(s) of the study at the end of your conclusion section.

 Ans: The statement was inserted in the final section of the conclusion. “However, sensory evaluation, which is an important quality feature, can be undertaken further in the future to ensure the effectiveness of this approach.”

Some specific comments on the manuscript:

  1. Line 72: How do authors confirm that salted egg is a major egg product? Any statistical data to support this claim?

Ans: The phrase "As a major egg product" was removed to avoid overstatement.

  1. Line 141-143: The preparation for salted eggs using ozonised brine salting is unclear. Do you mean that the ozone generator is only used to prepare ozonised water? The ozonised water is then used to dissolve the salt, and the ozonised brine is used to prepare the salted eggs. The ozone generator is not used throughout the salting duration (15 days). Please ensure the preparation method is explained in detail so that readers can reproduce the method.

Ans: It was clarified that "The ozone generator was only used to prepare ozonized water. The ozonized water was then used to dissolve the salt, and the ozonized brine was used to produce the salted eggs." This statement has already been inserted into the text.3.     

Line 154: AgNO3

Ans: Done.

  1. Eq. 1: Please check your equation. Also, indicate the meaning of “5.8” in the equation.

Ans: Eq.1 was double-checked. In addition, "5.8" in the equation is defined as "a conversion factor". Actually, this is a typical equation to calculate the salt content of food samples using the titration method.

  1. Fig. 1: Please provide statistical analyses on the data points. Are they significantly different?

Ans: Done.

  1. Line 413: Please check the superscript.

Ans: Done.

  1. Fig. 7: Authors should also show the salted egg yolk’s cross-section or interior appearance. Solidification starts from the outer layer. Hence, showing the cross-section of the salted egg yolk is an important criterion to determine the salted egg’s readiness. 

Ans: Thank you very much for your helpful guidance. In our next study, we will show the cross-section or internal appearance to indicate solidification and salted egg readiness. However, in this study, we compare the exterior appearance of unboiled salted egg to fresh egg, and it is clear that the shape and color of the yolk and albumen have been altered.

  1. Line 533: Without sensory evaluation, the acceptance is merely related to visual appearance. Please indicate this in the manuscript.

Ans: This statement was removed because sensory evaluation was not performed in this study. Thank you very much.

Reviewer 2 Report

Manuscript foods-2419842, entitled “A Novel Approach for the Production of Mildly Salted Duck Egg Using Ozonized Brine Salting”

General comment

This research article provides useful information about the effects of using ozonized brine salting on the characteristics of duck eggs. Although, it is in general appropriately organized and written, there are some points that should be corrected or clarified.

L15-16: “This method, however, induces a high salt content in the product, which may be linked…”

L27: “levels” instead of “readings”

L30: “…both salted yolks showed increased TBARS values after cooking…”

L41: “rich” instead of “high”. Please delete “all”

L42: Please explain the abbreviation “PDCAAS”

L47: “deterioration” instead of “loss”

L52-53: “…solutions to preserve them, maintain their nutritional value and extend their shelf-life [6].”

L85: “…expensive to be produced…”

L99: “result in improved” instead of “assisted to maintain”

L111-112: “result in protein oxidation” instead of “produce oxidative stress in proteins”

L113-115: Please delete (repetition)

L117: “…following ozonation [24] At the same time, the changes…”

L121: “was increased” instead of “grew”

L127: “…eggs by utilizing…”

L139: “allocated” instead of “divided”

L188: “is presented” instead of “was stated”

L252: “This finding could be attributed to the fact that the ozonized…”

L254-255: “was more evident” instead of “can be especially seen”

L267: “produced” instead of “made”

Figure 1: Please indicate significant differences (superscripts)

L275: “several” instead of “the following”

L276: “functionality”

L280: “This effect was due to…”

Figures and text: Please use a uniform characterization of groups. CB or Brine or Traditional Salting? OB or Ozonized Brine or Ozonized Salting?

L289: “provides” instead of “gives”

L290: What do you mean by “the salted egg is ready to be harvested when the yolk index is close to 1.0.”? How is this sentence related with your findings?

L307: “Figure 3a illustrates the pH values of fresh…”

L318: Please delete “reportedly”

L342: “…characteristics can induce egg protein and lipid oxidation. Figures 3a…”

L343: “show” instead of “depict”

L350: “…proteins that served as a barrier…”

L359-360: “…undergo more extensive modifications during…”

L379: “To examine the extent of lipid oxidation of salted…”

L383: The study of Larouche et al. [42] refers to eggs?

L394: “caused” instead of “brought on”

L399-400: How did you reach to this conclusion? High TBARS values also in yolk.

L405: “induced” instead of “brought on”

L464 and throughout the text: In general attribute L* refers to lightness or brightness and not to whiteness

L467: “similar” instead of “comparable”

L476-477: “…index of yolk was similar in ozonized and traditional brine.”

L503: “As indicated” instead of “From the results”

L512: Please delete “lower”

L513: “caused” instead of “brought on”

L517-518: “…less than these with conventional brine. This fact demonstrated…”

L524-525: Please delete “within the diet”

L532: “According to our results, ozonized…”

Moderate editing of English language required

Author Response

Reviewer#2

Manuscript foods-2419842, entitled “A Novel Approach for the Production of Mildly Salted Duck Egg Using Ozonized Brine Salting”

General comment

This research article provides useful information about the effects of using ozonized brine salting on the characteristics of duck eggs. Although, it is in general appropriately organized and written, there are some points that should be corrected or clarified.

L15-16: “This method, however, induces a high salt content in the product, which may be linked…”

Ans: Done.

L27: “levels” instead of “readings”

Ans: Done.

L30: “…both salted yolks showed increased TBARS values after cooking…”

Ans: Done.

L41: “rich” instead of “high”. Please delete “all”

Ans: Done.

L42: Please explain the abbreviation “PDCAAS”

Ans: Done.

L47: “deterioration” instead of “loss”

Ans: Done.

L52-53: “…solutions to preserve them, maintain their nutritional value and extend their shelf-life [6].”

Ans: Done.

L85: “…expensive to be produced…”

Ans: Done.

L99: “result in improved” instead of “assisted to maintain”

Ans: Done.

L111-112: “result in protein oxidation” instead of “produce oxidative stress in proteins”

Ans: Done.

L113-115: Please delete (repetition)

Ans: Done.

L117: “…following ozonation [24] At the same time, the changes…”

Ans: Done.

L121: “was increased” instead of “grew”

Ans: Done.

L127: “…eggs by utilizing…”

Ans: Done.

L139: “allocated” instead of “divided”

Ans: Done.

L188: “is presented” instead of “was stated”

Ans: Done.

L252: “This finding could be attributed to the fact that the ozonized…”

Ans: Done.

L254-255: “was more evident” instead of “can be especially seen”

Ans: Done.

L267: “produced” instead of “made”

Ans: Done.

Figure 1: Please indicate significant differences (superscripts)

Ans: Done.

L275: “several” instead of “the following”

Ans: Done.

L276: “functionality”

Ans: Done.

L280: “This effect was due to…”

Ans: Done.

Figures and text: Please use a uniform characterization of groups. CB or Brine or Traditional Salting? OB or Ozonized Brine or Ozonized Salting?

Ans: The terms “brine” and “ozonized brine” were used throughout the text.

L289: “provides” instead of “gives”

Ans: Done.

L290: What do you mean by “the salted egg is ready to be harvested when the yolk index is close to 1.0.”? How is this sentence related with your findings?

Ans: It means on its own because it has been stated that when the yolk index approaches 1.0, the salted egg is ready to be harvested. As a result, the sentence was rewritten. “In this experiment, the ozonized brine salted egg had a slightly higher yolk index (0.62) than the brine salted egg (0.55) and fresh egg (0.47), respectively (p < 0.05; Fig. 2b). As a result, ozonized brine salted eggs had greater yolk index quality than brine salted eggs, and they may be ready to harvest sooner.

L307: “Figure 3a illustrates the pH values of fresh…”

Ans: Done.

L318: Please delete “reportedly”

Ans: Done.

L342: “…characteristics can induce egg protein and lipid oxidation. Figures 3a…”

Ans: Done.

L343: “show” instead of “depict”

Ans: Done.

L350: “…proteins that served as a barrier…”

Ans: Done.

L359-360: “…undergo more extensive modifications during…”

Ans: Done.

L379: “To examine the extent of lipid oxidation of salted…”

Ans: Done.

L383: The study of Larouche et al. [42] refers to eggs?

Ans: It does not refer to eggs. These TBARS values are generally recommended for all lipid-based products. So, it was changed to “In general, lipids can be classified as not oxidized (TBARS value < 1.5 mg MDA/kg), moderately oxidized (1.6 < TBARS value < 3.6), or oxidized (TBARS value > 3.7), according to Larouche et al. [42].”

L394: “caused” instead of “brought on”

Ans: Done.

L399-400: How did you reach to this conclusion? High TBARS values also in yolk.

Ans: The statement was changed. “It was claimed that ozonized brine appeared to induce protein oxidation to a desired degree.

L405: “induced” instead of “brought on”

Ans: Done.

L464 and throughout the text: In general attribute L* refers to lightness or brightness and not to whiteness

Ans: In this study the “whiteness” of the boiled salted egg albumen was calculated from the L*,a*, and b* values. The method for whiteness determination was stated in Section 2.3.6.

L467: “similar” instead of “comparable”

Ans: Done.

L476-477: “…index of yolk was similar in ozonized and traditional brine.”

Ans: Done.

L503: “As indicated” instead of “From the results”

Ans: Done.

L512: Please delete “lower”

Ans: Done.

L513: “caused” instead of “brought on”

Ans: Done.

L517-518: “…less than these with conventional brine. This fact demonstrated…”

Ans: Done.

L524-525: Please delete “within the diet”

Ans: Done.

L532: “According to our results, ozonized…”

Ans: Done.
